# Public decisions about COVID-19 vaccines: A UK-based qualitative study

**Simon N. Williams**[1,2]*, **Christopher J. Armitage**[3,4,5], **Kimberly Dienes**[1,3], **John Drury**[6], **Tova Tampe**[7]

**1** School of Psychology, Swansea University, Swansea, Wales, United Kingdom, **2** Department of Medical Social Sciences, Feinberg School of Medicine, Northwestern University, Chicago, Illinois, United States of America, **3** Manchester Centre for Health Psychology, University of Manchester, Manchester, United Kingdom, **4** Manchester University NHS Foundation Trust, Manchester Academic Health Science Centre, Manchester, United Kingdom, **5** NIHR Greater Manchester Patient Safety Translational Research Centre, University of Manchester, Manchester, United Kingdom, **6** University of Sussex, School of Psychology, Falmer, United Kingdom, **7** Independent Researcher, Kassel, Germany

* s.n.williams@swansea.ac.uk

**Data Availability Statement:** Ethics was approved by Swansea University's Department of Psychology Ethics Committee. As part of the ethics review process, participant confidentiality restrictions prohibit the authors from making the

## Abstract

### Objective

To explore UK public decisions around whether or not to get COVID-19 vaccines, and the facilitators and barriers behind participants' decisions.

### Design

This qualitative study consisted of six online focus groups conducted between 15[th] March and 22[nd] April 2021. Data were analysed using a framework approach.

### Setting

Focus groups took place via online videoconferencing (Zoom).

### Participants

Participants (n = 29) were a diverse group (by ethnicity, age and gender) UK residents aged 18 years and older.

### Results

We used the World Health Organization's vaccine hesitancy continuum model to look for, and explore, three main types of decisions related to COVID-19 vaccines: vaccine acceptance, vaccine refusal and vaccine hesitancy (or vaccine delay). Two reasons for vaccine delay were identified: delay due to a perceived need for more information and delay until vaccine was "required" in the future. Nine themes were identified: three main facilitators (Vaccination as a social norm; Vaccination as a necessity; Trust in science) and six main barriers (Preference for "natural immunity"; Concerns over possible side effects; Perceived lack of information; Distrust in government;; Conspiracy theories; "Covid echo chambers") to vaccine uptake.

data set publicly available. During the consent process, participants were explicitly guaranteed that the data would only be seen my members of the study team. For any discussions about the data set please contact Swansea University's Research Governance: resgov@swansea.ac.uk.

**Funding:** This research was supported by the Manchester Centre for Health Psychology based at the University of Manchester (£2000) and Swansea University's 'Greatest Need Fund' (£3000). This research was supported by the Manchester Centre for Health Psychology based at the University of Manchester (£2000) and Swansea University's 'Greatest Need Fund' (£3000). The funders had no role in study design, data collection and analysis, decision to publish, or preparation of the manuscript.

**Competing interests:** CJA is supported by NIHR Manchester Biomedical Research Centre and NIHR Greater Manchester Patient Safety Translational Research Centre. This does not alter our adherence to PLOS ONE policies on sharing data and materials. JD sits on SAGE SPI-B subgroup, and Independent Sage. TT currently works for the World Health Organization, but contributed to this paper as an independent researcher. This does not alter our adherence to PLOS ONE policies on sharing data and materials." (as detailed online in our guide for authors http://journals.plos.org/plosone/s/competing-interests The authors have no other relationships or activities that could appear to have influenced the submitted work.

## Conclusion

In order to address vaccine uptake and vaccine hesitancy, it is useful to understand the reasons behind people's decisions to accept or refuse an offer of a vaccine, and to listen to them and engage with, rather than dismiss, these reasons. Those working in public health or health communication around vaccines, including COVID-19 vaccines, in and beyond the UK, might benefit from incorporating the facilitators and barriers found in this study.

## Introduction

Vaccine hesitancy is a complex and multifaceted problem, and one that is influenced by a range of contextual (e.g. historical, institutional, political) factors, as well as individual-level and vaccine specific factors (e.g. costs or design of a given vaccination program) [1]. Individual-level factors, include health-system and providers, knowledge and beliefs about health and prevention, personal perceptions about risk versus benefit and personal and family experiences with vaccination (including pain and side effects from past vaccines) [1, 2]. Although they are shaped by contextual factors, this research is primarily interested in individual perceptions of UK residents around the decision to get vaccinated against COVID-19.

Vaccine hesitancy can be defined as "the delay in acceptance or refusal of vaccination despite availability of vaccination services" [2]. In this paper, we draw on The World Health Organization's (WHO) SAGE Working Group on Vaccine Hesitancy's 'continuum of vaccine hesitancy' model, which sees vaccine views to be set on a continuum between full acceptance of vaccines with no doubts, through to complete refusal with no doubts [1, 2]. Vaccine hesitancy is seen as a heterogenous group in-between these diametric positions, including those who "delay" acceptance (i.e. do not get it when first offered or according to schedule).

Although the WHO continuum of vaccine hesitancy is a simple and useful heuristic to categorise individuals, especially in relation to individual-level perceptions or attitudes, it is important to acknowledge that these attitudes do not occur in a vacuum but are shaped and influenced by many social and contextual factors. Intentions and decisions around vaccine acceptance or refusal, and the reasons behind them, can also be understood in terms of the '5 C's' model of vaccine hesitancy. This model suggests that vaccine hesitancy is influenced by: *Complacency*—for example where the perceived risk of harm from the disease is low; *convenience*—for example where there are practical or logistical barriers to access vaccination; *confidence*—for example where there is a lack of trust in the safety or efficacy of the vaccine; *communication*–for example where misinformation can create distrust or confusion; and, *context*–for example certain social groups, including some ethnic minority communities might encounter additional barriers, including structural racism, which might affect vaccine uptake [1, 3]. Another theoretical framework is the Behavioural and Social Drivers (BeSD) of vaccine uptake [4]. The BEsD framework suggests four main drivers of vaccination uptake: (1) people's mental and emotional responses to vaccines (*thinking and feeling*); (2) social or group norms around vaccinations (*social processes*); (3) people's willingness and intentions, or hesitancy, to get vaccinated (*motivation*); (4) contextual or structural barriers related to e.g costs or access (*practical issues*) [4]. It is important to note the inter-relatedness of the many drivers of vaccine acceptance or hesitancy. As such, a focus on individual level decisions or intentions around vaccines, as is the case in this study, needs to acknowledge the ways in which individual feelings, beliefs and motivations, are shaped by (and serve to shape) contextual and practical issues.

In terms of the reasons behind vaccine intentions and decisions, survey data on COVID-19 vaccine intentions and decisions suggests that most common reasons for vaccine hesitancy include: worries over side effects, worries over long term effects on health, as well as concerns over its efficacy [5]. Qualitative research on public views on COVID-19 vaccines is emerging. One study, from the UK, found that vaccine hesitancy was associated with three main factors: safety concerns, negative stories and personal knowledge, with those who were most confused, worried and mistrusting being the most hesitant [6]. Another study, from Canada, on overall attitudes to public health measures to reduce COVID-19 transmission found that many participants felt that vaccines were a means to "get back to normal life" while some were hesitant due to a lack of confidence in the potential efficacy of the vaccine and concerns over side-effects [7]. A study from Australia, with hesitant health or social care workers or clinically vulnerable adults, found that participants saw vaccination as beneficial for both individual and community protection, but also expressed safety concerns that made them feel like "guinea pigs" [8].

Ongoing research into vaccine hesitancy is needed to follow how attitudes and decisions around COVID-19 vaccines may be changing as the pandemic continues. Also, qualitative data can explore, in depth, the reasons behind why people are deciding to get vaccinated or not. In this paper, we explore participants' decisions on COVID-19 vaccines in the UK during March and April 2021. For context, during this period the UK was experiencing a rapid roll out of COVID-19 vaccines (Astra Zeneca and Pfizer-BioNTech) via the National Health Service, with between January and 22nd April 2021, administered approximately 35 million total doses, with approximately 60% of the total population aged 16 and over having received at least one dose with doses being prioritised amongst older adults and those with certain underlying health conditions (clinically vulnerable and clinically extremely vulnerable adults) [9].

This paper explores participants' intentions and decisions around whether or not to get vaccinated, and specifically the reasons behind them, thereby contributing to our understanding of the facilitators and barriers to vaccine uptake.

## Materials and methods

### Participants and data collection

Data from this study came from the COVID Public Views (PVCOVID) study–a mixed-methods study using panel focus groups and surveys during the pandemic (commenced March 2020) [10, 11]. Participants for the PVCOVID study were initially recruited to the study from March-July 2020, with a total of 53 participants initially enrolling into the study. Participants were all UK-based adults aged 18 years or older. Recruitment for the study took place primarily via non-probability, opportunity sampling. Recruitment included using social media advertising (Facebook ads and via posting ads on Twitter), other online advertising (e.g. online 'free-ads' such as Gumtree), as well as snowball sampling (e.g. asking participants who had taken part in a focus group to distribute the study ad to others they felt might be interested in participating). Recruitment sought as diverse a range of ages, genders, race/ethnicities, UK locations, and social backgrounds as possible (e.g. advertisements encouraged expressions of interest from individuals from Black and Asian Minority Ethnic (BAME) backgrounds; social media ads were designed to targeted users from across the UK and a wide age range). Although the study had a low number of individuals from older age groups (over 50 years of age) the over-representation of younger adults could be seen as beneficial because of their lower vaccination coverage [9].

Here we report on data from six online focus groups with 29 participants from within the overall PVCOVID study. Focus groups were not arranged according to any pre-existing views or decisions around vaccinations (i.e. we did not purposefully put those who were not

intending on getting vaccinated in the same group for example). This was largely because of the longitudinal nature of the overall study, and our initial decision to try to keep the membership of each focus group the same over time in order to build rapport, familiarity and openness within the groups (we did not collect data on vaccine views during the initial recruitment and group allocation in March 2020). Each group contained a mixture of those who had already received at least one vaccine (n = 15) and those who had either already refused a vaccine or who were delaying their decision to get a vaccine (n = 14). One potential limitation of this focus group composition was that those who were refusing or delaying vaccination might have felt less comfortable expressing their opinion (since, generally getting vaccinated was seen by many as a social norm—see below). However, the focus group facilitator sought to ensure that all participants felt comfortable expressing their views, and that dialogue within the group was not hostile and as respectful and open as possible. Also, questions were phrased in non-leading, non-judgemental a way as possible.

In March 2021, participants were invited to take part in a rapid round of focus groups on the topic of vaccines. Participants took part in focus groups conducted between 15th March and 22nd April 2021. Further information about the participants discussed in the present paper are presented in Table 1.

Online focus groups were necessary due to COVID-19 social distancing regulations, but have been seen to have benefits in general, as a means of eliciting public views from diverse and geographically dispersed participants [12, 13]. Each focus group (of 4–6 participants) met virtually via the videoconferencing platform Zoom for approximately one hour. All focus groups discussed in the present paper were moderated by SW. Focus groups were recoded and transcribed. The topic guide for the focus groups (Appendix 1) was initially developed using existing literature on vaccination public attitudes and vaccine hesitancy discussed above, as well as rapidly emerging surveys on public attitudes to COVID-19 public attitudes.

Ethical approval was received by Swansea University's School of Management Research Ethics Committee and Swansea University's Department of Psychology Ethics Committee (Ref: 2020-4952-3957). All participants gave informed consent, both written and verbal. All data were kept securely and confidentially in line with ethical requirements, and where data is presented below, all quotes are anonymised to protect participants' identities.

## Analysis

Data were analysed in accordance with a Framework Analysis (FA) approach [14]. FA is a flexible approach that is not aligned with a particular epistemological, philosophical or theoretical

**Table 1. Demographic details reported by participants.**

| Characteristic | N (%) |
|---|---|
| *Gender* | |
| Female | 11 (38) |
| Male | 18 (62) |
| *Age range* | |
| 20s | 9 (31) |
| 30s | 8 (28) |
| 40s | 9 (31) |
| 50+ | 3 (10) |
| *Ethnicity* | |
| White | 20 (69) |
| BAME | 9 (31) |

tradition. It can be either primarily inductive or deductive (or a combination thereof) and can be adapted with many qualitative approaches with the main aim being to generate themes [14]. Our use of FA combined elements of both induction and deduction in an abductive approach [15, 16], whereby the researchers inductively coded for emergent facilitators and barriers to vaccine uptake as they emerged. The coding process was also broadly informed deductively by existing literature, including the WHO's 'continuum of vaccine hesitancy' framework. Using the vaccine hesitancy continuum model [1], we analysed data to look for, and explore, three main types of vaccine decisions in the data: those who had accepted, or were planning on accepting, the vaccine; those who had refused, or were planning on refusing, the vaccine; and those who had not yet decided, or were delaying the decision of, whether or not to get the vaccine. As such, each participant was coded into one of these three main categories based on their decision or intention (e.g. statements that indicated they were not sure if they wanted a vaccine were coded into three main categories: accept, delay, refuse. Sub-coding sought to bring out the nuances within these simplified intention/decision types (e.g. "accept but unsure"). Inductive analysis was used to explore facilitators and barriers to vaccine acceptance (i.e. the reasons why people were getting or intending on accepting or refusing a vaccine or why they were unsure of or delaying their decision).

Two authors (SW and KD) primarily analysed the data, in consultation with the other three authors (CA, JD, TT). The first stage of FA is *data familiarisation*. SW and KD independently read three focus group transcripts initially, and independently assigned preliminary codes to specific lines or chunks of texts. The two authors met regularly (after each transcript) to identify commonalities and differences in their coding and discussed frequent or significant codes which were beginning to form emergent themes–i.e. patterns in the data, codes clustering around a common concept (e.g. 'preference for natural immunity') [16, 17]. Following this, the third stage–*data indexing*–was performed, through which the two analysts systematically applied the thematic framework and the existing list of codes to the remaining transcripts. As FA is a flexible, iterative approach, new codes, and new themes (clusters of codes) were developed as they emerged. The fourth stage, *data charting*, enabled the analysts to refine, rearrange and focus (data reduction) on what they deemed the most prevalent and salient themes. The final stage–*data mapping*–enabled the analysts to review the main themes and organise them in relation to the study's main objective–identifying facilitators and barriers to COVID-19 vaccine uptake using the accounts of the participants intentions and decisions around vaccines. The other three authors (CA, JD, TT) were not involved primarily in the data analysis but were consulted throughout the FA–and provided input into code definition and theme generation, they also provided comments which were used to resolve any disagreements between SW and KD's coding.

## Results

### Vaccine acceptance

One type of response, those who had already accepted or planned to accept (when they were offered one), a COVID-19 vaccine, held largely positive views around COVID-19 vaccines. In line with the vaccine hesitancy continuum [1], those *accepting* the vaccines fell into two subtypes: those who expressed little to no reservation about their decision to accept a vaccine ("full acceptance") and those who had accepted, or were intending to accept, a vaccine, but who also expressed concerns or reservations about their decision ("accept but unsure").

Those fully accepting tended to frame their decision as something that was quite "normal" to them ("I've had all my vaccines"—see *vaccination as a social norm* below):

"I had no problems getting the vaccines or any concerns about the efficacy of the vaccines . . . It was something I need to do to protect my family and loved ones . . . It was an easy decision to make, I've had all my vaccines all my life, because I see the value." (Participant 44, Female 30s)

Those fully accepting also framed vaccination as a collective responsibility, and their decision as something that would "protect" others as well as themselves (especially vulnerable others) ("I don't want to hurt anyone else" (Participant 19, Male, 20s)). This included some who saw themselves as not necessarily being at risk personally:

"I had my first one [vaccine dose] about a month ago . . . I was practically queuing up as soon as I heard . . . My biggest thing throughout the whole pandemic to protect others, not necessarily myself, because I've never seen myself as being particularly at risk but now, I am statistically less likely to catch and pass it on" (Participant 6, Male, 20s)

However, although some of those accepting were confident in their decision, others expressed concerns in spite of ultimately deciding to get vaccinated (akin to the Vaccine Hesitancy Continuum's 'accept but unsure' vaccine attitude group) [1, 2]:

"I was a bit suspicious. I suppose it's only natural, everyone's a little worried because you think it's only been done in a year, but the science today compared to years ago is outstanding isn't it, so they have more methods of testing to see if its effective, so I wasn't really worried, and I'm glad I have got it" (Participant 42, Female 30s)

Those accepting a vaccine—including those in the "accept but unsure" category–expressed and emphasised a number of facilitators, including a *trust in science* and *vaccination as a necessity* (see below).

### Vaccine refusal

A second type of response was from those who had already refused, or who were planning on refusing, a COVID-19 vaccine. Unlike those who were unsure of, or who were delaying, their decision of whether or not to get a vaccine (see below), those refusing explicitly stated that they were not going to have the vaccine (at least not for the "foreseeable future"):

"I've not had the vaccine and, to be honest with you, I don't have any intention of getting the vaccine like in the foreseeable future, I just don't see the point". (Participant 3, Male, 20s)

Those refusing the vaccine tended to frame vaccination in terms of individual rather than collective responsibility; and as a personal choice ("It's a choice I've made" (Participant 8, Female, 40s)). Those refusing were also less likely than others to frame their vaccination as something that would protect others and saw it as something that was primarily useful for those most vulnerable from serious COVID-19-related illness, for example older age groups (see *preference for natural immunity* below). Those refusing also emphasised the three main barriers, including a *preference for natural immunity*, *concerns over side effects* and *distrust in government* (see below).

### Vaccine delay

A third type of response was from those who were still unsure of, or were delaying, the decision of whether or not to get the vaccine. These participants can be distinguished from those who had already made up their mind that they did not want to get vaccinated. These participants

felt they needed more time to decide. Some of those delaying had already turned down or ignored their vaccination appointment invitations. However, they characterized their current position as subject to change–that is they did not want the vaccine *yet* but were aware that they *might* in future:

"I wouldn't take it straight away. . . I'd rather just wait and make my own decision at my own pace really" (Participant 15, Male, 20s)

Some of those delaying tended to emphasise a need for more information about the efficacy and safety of the vaccine as a main reason for the delay in their decision (see *perceived lack of information* below) with some feeling they were being "pressured" (e.g. by the vaccination campaign and offer) into having to make a decision too quickly:

"I'm still holding out. I've been sent several letters and loads of text messages telling me that I needed to go for the vaccination, but I would rather wait at least four months to see exactly where it is all going. . . I don't want to be sort of pressured into something that I might regret" (Participant 15, Male, 40s)

Like those refusing the vaccine, those delaying their decision were also concerned about the speed at which vaccines had been developed and were concerned over the possible longer-term side effects (see *concerns over possible side effects* below):

"I am a little bit nervous because I do feel like it's all been rushed through incredibly quickly . . . have they done all the tests? I wanted to ensure that a lot more people had it before it was my turn, because I kind of figured that if there's a problem, we'll know" (Participant 50, Female, 50+)

Despite the feeling of "pressure", some participants delaying tended to perceive themselves as having agency in their choice of whether to get vaccinated. However, others delaying tended to argue they felt as though they would be "forced to do it" (Participant 7, Female 20s):

"I don't intend on taking it, unless it's a must, unless, I heard you have [a vaccine] to go on holidays, but right now I am going to refuse it, until the very last moment. But I know at some point I will have to take it" (Participant 18, Female, 30s)

These participants tended to be quite hesitant about the idea of getting a COVID-19 vaccine but believed that they would "have" to get one in order to be able to engage in certain activities in future, particularly traveling internationally. Here, vaccination tended to be viewed as a personal choice rather than in terms of collective responsibility ("I have the say and right what I should accept into my body and nobody else should have a say" (Participant 22, Female, 20s)). They tended to perceive themselves as having little or no agency in regard to whether or not they would ultimately have a vaccine. The act of delaying may therefore be construed as a way to preserve agency while they could, or perhaps to recapture a sense of control over their lives that many felt they had lost during the pandemic [10].

## Facilitators: Reasons for vaccine acceptance

**Vaccination as a social norm.** Social norms are a means through which health behaviour and decisions can be influenced during the pandemic [18]. Those accepting a vaccine tended to discuss COVID-19 vaccination in relation to a personal willingness to be vaccinated in the

past and in general ("I've had vaccines all my life for what is needed" (Participant 12, Male, 40s)), as well as in relation to a culture (the UK) within which vaccines were normative (including the fact that vaccinations were a "part of normal life" from a young age):

"For me, I've never been one to say no to a vaccine. Its part and parcel—as a child you are given various vaccines, and you don't necessarily have a choice, but then when you have your boosters, you just go ahead and do it and to me that's part of normal life" (Participant 10, Male, 40s)

For some, vaccination for COVID-19 was becoming more normative over the course of the pandemic. For some, hesitancy was decreasing, as more people started to get vaccinated, partly as a way to help get back to "normality" (see *vaccination as a necessity*):

"Talking to people there's a lot of people have taken it [the vaccine] and I'm surprised. They had said initially that they weren't going to take it . . . [but] there's just a lot of impatience and frustration at the minute you know people just want to get back to some sort of normality" (Participant 8, Female, 40s)

Others felt as though there was a pressure to get vaccinated, for example within some workplaces:

"Initially when it [vaccine] came out a lot of the [front line-workers] on my team were like 'nope, not going to get it' but that changed one-by-one, they started to go for it, and now I think there is only a handful who haven't gone for it . . . its being monitored by senior management but they have got to ask have you had it or not, they are not asking why you haven't had it, but I don't know if that is something at some point, you know you have to justify your grounds as to why you haven't" (Participant 5, Female, 30s)

For this participant, in their workplace, there was an implied expectation that they would, in the future, need to get the vaccine or justify why they hadn't. This informal scrutiny of vaccine uptake may be functioning as an example of 'norm-nudging' [18]. In this participant's workplace, although vaccination was not obligatory or coercive, and the reasons for 'opting-out' were not required, the implication was that opting-in was the default or normative position. The expectation that justification for opting out would be necessary in the future may have led to the change in decision (i.e. individuals opting to get vaccinated now, despite not initially wanting to, because they felt they would either be required to in the future or would have to justify their decision not to).

**Vaccination as a necessity.** Vaccination as a social norm closely related to a second facilitator–vaccination a necessity. Participants discussed three main ways in which vaccination would be necessary or inevitable. These were framed either positively or negatively (related to whether participants were accepting of vaccines or not).

Firstly, those accepting a vaccine tended to refer to vaccination as being the way for them to "get back to normal" (Participant 20, Male, 20s) or "the only way forward, to get on with our lives" (Participant 12, Male, 40s). Secondly, a number of participants, including those accepting a vaccine and those delaying their decision, described an "acceptance" of the fact that they would need regular vaccinations or 'boosters' as a result of the continued emergence of new variants of the virus:

"My parents and I have already accepted that we will require boosters, they will be a fact of life, probably for the next four or five years, and it will be the same as our flu jab that we get every year" (Participant 2, Male, 40s)

In particular, concerns over new and emerging variants were cited as a reason as to why vaccination was not perceived to be a "one-off", and so "regular" vaccination was seen to be necessary with "no other way out".

"The virus is going to keep mutating, there is going to be different variants, I think that's why the government is so hot on this vaccine because it is not going to stop. . . . It's something that we're going to have to live with this is not going to die out anytime soon. . . there is no other way out . . . we are going to have to go for our top-ups with the vaccine" (Participant 5, Female, 30s)

It is worth noting however, that refusers saw the possibility of regular vaccination as a deterrent to getting vaccinated currently:

"There's a concern that, if one takes the vaccine have a variant of a new variant emerges. That could be resistant to the vaccine and then there's a question of do you have to keep taking vaccines to protect yourself against each single various oh I don't really have any intention of getting the vaccine" (Participant 3, Male, 20s)

Thirdly, immunity certificates or "vaccine passports" were also cited by some participants as a potential facilitator to vaccine uptake. One way in which they were framed was as a "necessary evil"—as something that were, begrudgingly, needed to help "save" the economy:

"I think it's just it's almost a necessary evil and it's the way we're going to go . . . I think it's also going to save a lot of businesses and that will save you know; the economy and we all need the economy to be boisterous we all need a prominent economy". (Participant 2, Male, 40s)

However, vaccine passports were a complex and controversial topic in the focus groups. Some delayers felt that vaccine passports were a reason why they would ultimately be "forced into" get a vaccine:

"I'm two minds really about whether to get the vaccine or not but I have a feeling people will be pressurised into having it through the vaccine passports or certificates, whether that could be for traveling or getting jobs, so I think people will be indirectly forced into getting it" (Participant 7, Male, 40s)

**Trust in vaccine science.**    Accepters expressed a higher trust in science, and to frame science as being relatively independent from government:

"I chose to be vaccinated because, not that I trust the government, but I trust the medicine and the science behind it. It's not the government that produces the vaccine." (Participant 11, Male, 50+)

Accepters were also more likely than others to associate the vaccination program with the health service (NHS):

"I know that the vaccine was developed really quickly . . . but I have faith in the health system and its testing" (Participant 10, Male, 40s)

Although some accepters expressed some concern at the speed at which the vaccines had been developed, they tended to contextualise this in terms of the fact that science was now more advanced than with many previous vaccines:

"I was a bit suspicious; I suppose it's only natural, everyone's a little worried because you think it's only been don't in a year, but the science today compared to years ago is outstanding isn't it, so they have more methods of testing to see if its effective, so I wasn't really worried, but I'm glad I have got it" (Participant 25, Female, 30s)

They also contextualised the speed of vaccine development in terms of what they perceived as rigorous testing, and the fact that there had been considerable scientific, medical and financial focus and investment in the vaccine development. This for them meant any potential risks from unforeseen issues due to safety or efficacy were likely to be minimal and outweighed by the benefits of the vaccination program:

"I believe it's been so rigorously tested and it's the only thing spoken about medically for the last 12 months and so the risk for me was just so minimal and benefits outweighed any risk for me" (Participant 6, Male, 20s)

### Barriers: Reasons for vaccine delay or refusal

**Preference for 'natural immunity'.**   Some participants argued that one of the reasons they were either hesitant about or did not want to get vaccinated was because they preferred to "fight" the virus "naturally":

"For me personally I am not sure I would go for the vaccine. . . I just hope I have a strong immune system so I can fight the virus. We have this in-built immune system within our bodies. . .give them a chance to operate" (Participant 7, Male 40s)

In discussing their decision, refusers tended to frame COVID-19 as a disease which tended not to affect young and "healthy" people:

"I don't have any intention of getting the vaccine in the foreseeable future, I just don't see the point, because the virus is mainly fatal to those who are, like middle aged. (Participant 3, Male, 20s)

In doing so, they also emphasised their own healthiness as a reason as to why they didn't need the vaccine, and drew comparisons to the fact they hadn't needed vaccines in the past for other diseases:

"It just does not make sense to me to take a vaccine, it's like a flu vaccine. I've never ever taken a flu vaccine, because I don't get the flu." (Participant 8, Female, 40s)

As noted above, refusers framed vaccination as an individual act rather than a collective act, and argued that the lack of personal benefit was outweighed by the perceived risks posed from potential side effects:

"I have no intentions of taking it, and I have focused a lot on my health over the years, I'm the healthiest I've ever been and I just don't see I don't the reason for me to take it . . . because from what I've read, there are risks with it so." (Participant 8, Female, 40s)

**Concerns over possible side effects.** One of the main reasons for vaccine hesitancy was a concern over side effects–something that accounted for why some people were delaying the decision to get vaccinated:

"I probably will have it [a vaccine], but I want to wait to see if people turn into zombies first. I'll wait until a few hundred thousand have had it first" (Participant 28, Female, 20s).

Although the above quote was tongue-in-cheek, it was indicative of a wider concern over potentially unforeseen, longer-term, side effects. Delayers tended to frame these concerns in relation to what they knew about how vaccines were "normally" developed. They discussed how comparatively quickly COVID-19 vaccines had been developed and emphasized how not enough time had passed to be able to know long-term side effects:

"I do have like some concerns about how quickly they developed this vaccine, because most vaccines take like you know six five to six years to test and to make sure that you know they've seen all the side effects. But with this vaccine I still have that reservation that maybe it's been too quick, and they've not really teased out all the long-term effects." (Participant 3, Male, 20s)

Other delayers focused on short-term side effects or risks. In particular, recent reports of potential blood clots linked to vaccines were cited by a number of delayers. Interestingly, these participants were aware that any causal link between blood clots and vaccines had not been clearly demonstrated, or that any potential risk between vaccines and blood clots was considerably small. Nevertheless, they continued to cite blood clots as a cause for concern and framed their hesitancy in relation to it as an example of potential side effects (including unknown side effects that may emerge in the future):

"Right now, I am going to refuse it, until the very last moment. I feel like I'm a guinea pig. I don't know if you heard the news that they have stopped one of the vaccines because there has [sic] been cases of blood clots of something. I know it's a very, very, very, very tiny percentage but I feel like if I wait till the very last more [information] can come out". (Participant 9, Female, 30s)

Some participants distinguished between vaccines, with those who did expressing personal concern, or observed concern in others, over the vaccine manufactured by the company Astra-Zeneca, which had been the focus of blood clot controversies in the media. For example, one participant, who was vaccine hesitant, stated that although they accepted the Pfizer vaccine (i.e. the vaccine manufactured by Pfizer-BioNTech), they would not have accepted the Astra-Zeneca vaccine:

"Now I've had it I feel okay about it, but I think that's because with the Pfizer it doesn't seem to be any negative reports, whereas with AstraZeneca there seems to be a lot of mixed communications, and I don't think there is a lot of fault with the vaccine, it's just I don't think the company is very good at kind of being truthful and that makes people a bit

doubtful . . . I don't think I would have done it [had the AstraZeneca vaccine] (Participant 24, Female, 50+).

Another participant described how they had heard of others specifically opting out of vaccination, once they discovered they were to get the AstraZeneca vaccine:

"My aunt had hers and she said there was a huge queue in the surgery. . . but every single person that was offered the one that begins with an A [AstraZeneca] were actively declining it and walking out and she witnessed in ten minutes about 15 people turning up, being told what they were getting and walking out" (Participant 21, Female, 20s)

On the other hand, accepters, including those who had taken the vaccine with reservations (cf. The Vaccine Hesitancy Continuum's [1, 2] 'accept but unsure' group), were much less concerned over both short- and long-term side effects. They framed the issue in terms of a cost-benefit analysis, where for them, the perceived benefits (e.g. reducing the risk of "getting long Covid") outweighed any potential risk of unknown, long-term side effects:

"I'm quite worried about, not any side effects *now*, but like maybe in ten years' time . . . but then it seems like the risk from the vaccine is less than the risk of say getting long Covid . . . so the vaccine is the lesser of two evils" (Participant 24, Female, 50+)

They also made comparisons to what they saw as other equally rare side effects of common medications:

"I know there is this whole issue around blood clots, but people really need to get a grip, because you know, people die on a yearly basis from taking paracetamol, plenty of women die from blood clots as well . . .from taking the pill . . . that so the benefits outweigh the risks most definitely" (Participant 12, Male, 40s)

**Perceived lack of information.** A perceived lack of information was a major factor for why some participants were either refusing or delaying vaccination:

"Whatever is going on with the vaccine, I don't know, it really is a minefield of information" (Participant 8, Female, 40s)

Questions were scientific in nature and stemmed from the fact that COVID-19 was still such a new disease. For example, one participant (a refuser) questioned whether they would need to "keep taking" vaccines due to new variants (potentially "resistant" to the previous vaccines), something that deterred them from accepting their initial vaccine offer:

"There's a concern that, if one takes the vaccine and a variant of a new variant emerges that could be resistant to the vaccine then there's a question of do you have to keep taking vaccines to protect yourself against each single variant? I don't really have any intention of getting the vaccine." (Participant 3, Male, 20s)

Another participant (a delayer) questioned whether people still "need" a vaccine following infection with COVID-19:

"I am not against the vaccine, [but] for me there are so many unknowns, because It is so new there are so many questions that I want to ask, like if you have had Covid do you still

need, in terms of the antibodies you have, or do you still need the protection from the vaccine? I'm very much in the middle, so once I get those answers, I will be leaning more towards getting it" (Participant 5, Female, 30s)

**Distrust in government.** Refusers and some delayers tended to have less confidence in vaccine science and less trust in government ("I think a lot of my concerns are because of the government because I just don't trust them at all" (Participant 29, Female, 50+)). They framed COVID-19 vaccine science as being closely linked to, or even compromised by, political or economic interests. Some justified their distrust in relation to historical controversies or examples of medical iatrogenesis:

"The government doesn't have a very good track record with the sense that there was the Thalidomide tablets that were given to pregnant women back in the '60s, the blood transfusions that were imported from people volunteers in state penitentiaries in America that were contaminated, brought into the United Kingdom.. . . there's been several vaccinations given to toddlers . . . that came from America—I think about 1000 children have died.. . . I'm not one to trust governments, they tend to rush into things." (Participant 15, Male, 40s)

Although delayers generally recognised that the vaccines had been tested, they remained concerned or "sceptical" that testing had been done as extensively or for as long as was necessary. They tended to see vaccination as something that was still being tested (in the community):

"Even though there has been a lot of testing done, I still feel sceptical and quite scared to get it . . . technically it's still in the testing phases even though it's been approved, and so until I'm actually forced to do it, I don't think I want to" (Participant 7, Female, 20s).

Although, as discussed above, some participants framed vaccine passports as a "necessary evil" to enable them to travel or to help the economy, others, including both refusers, framed them as "Orwellian" and argued that by using them, government were encroaching on their "freedom of choice" (Participant 13, Male, 40s) and "privacy":

"[I'm] a hundred per cent against vaccine passports, I personally would rather just have the PCR [COVID-19 polymerase chain reaction] tests. . . . it is somewhat Orwellian . . . I 'm very, very concerned about things like surveillance and privacy" (Participant 3, Male, 20s)

**Conspiracy theories and misinformation.** Those refusing a vaccine, and some of those delaying, linked their distrust in government to what they saw as conspiracies (things that "don't seem to add up"), such as the perception that vaccines were driven by the "agenda" of the pharmaceutical companies involved in manufacturing them, in order to make profit.

"I mean distrust in government . . . the things that don't seem to add up. I mean we have got the pharmaceutical companies, several of them creating a vaccine, some kind of race . . . and it's just a win-win for them, if just everyone gets a vaccine and people can't think for themselves . . . it a big agenda" (Participant 7, Male, 40s)

It is important to emphasise that, to the participants themselves, these were seen as plausible conspiracies.

Amongst the Black and Asian Minority Ethnic (BAME) participants in this study, most were critical of the circulation of conspiracy theories. However, many of them did also discuss how conspiracy theories and misinformation was quite prevalent in their communities ("It's weird how it afflicts the Black community in terms of social media and WhatsApp conspiracy theories in circulation" (Participant 2, Male, 40s)). Some related the lower uptake in vaccination amongst BAME groups to the existence of 'folk wisdom' about what might help promote health or even protect against COVID-19:

"A few months back India was number two in the number of cases and deaths from Covid and one fine day it just vanished. So, everyone is trying to ask what is being done differently in India. . . and I guess that mindset is being transferred to racial communities here [in the UK]. So people are discussing our [Indian] food habits, and we do eat a lot of spicy food and spices so I've actually seen people talking about saying that 'ok it's our food which is different. . . There are people even saying 'avoid the vaccines, stick to your spices, your curries and you'll be fine'. So that is worrying but that is a topic which is being widely discussed in our Asian community" (Participant 4, Male, 30s)

However, a number of BAME participants linked the lower uptake in their communities to a distrust in government (see above) and thus a distrust in the vaccines. One example was the rumour that vaccines were being "tested" for side effects first amongst BAME patients:

"I think just from my experience, there's a lot of conspiracies that I've heard about. Because I mean I identify myself as a Black British person and so within my community I've heard a lot of just not trusting the vaccine . . . somebody sent me a video about [UK Government Health Secretary] Matt Hancock . . .suggesting that the vaccine was tested first amongst the BAME group." (Participant 5, Female)

These participants also discussed the issue of distrust as a wider issue, accounting for why there "is some cynicism in these communities" (Participant 2, Male, 40s). This lack of trust was seen to stem from a lack of information (cf. *lack of information* above), which in turn was seen to be the result of a lack of engagement with BAME communities, as well as a perceived lack of government accountability for the disparities that BAME communities have experienced during the pandemic:

"I just think within our community there needs to be a lot more education, especially if there are a lot of unknown questions that haven't been answered . . . that needs to be advertised more as to where we can go to ask those questions to be more equipped with the knowledge around these vaccines, rather than listen to these conspiracies which a lot of them is [*sic*] fake news. . .. A lot of it [lack of trust] stems from the government; a lot of how it [pandemic] has been handled is embarrassing, a high number of deaths were from the Black and Asian community and so that mistrust in government along with them not really putting their hands up just makes us even more anxious" (Participant 5, Female)

**Covid 'echo chambers'.** One potential barrier to uptake is what might be seen as the emergence of 'Covid echo-chambers'. Echo chambers can be defined as "polarized communities populated by like-minded" others and can be found particularly in online settings [19]:

"I also interact quite a lot on the internet and a lot of people that I speak to on the internet, they say the same thing that you know they just have this concern about the vaccine." (Participant 3, Male, 20s)

The existence of polarized views, on an emotionally charged subject led some participants to argue that "everyone is extreme in their reactions" (Participant 26, Female, 20s):

"It really divides those who do have the vaccine and those who don't, and I see quite a lot on social media people who post like very critical things of people who are on the opposite side" (Participant 15, Male, 20s)

As a result, some delayers felt 'as though they didn't 'fit' into either of the polarized attitude groups on vaccines, which may have been contributing to their uncertainty or hesitancy around whether or not to get vaccinated:

"Its either the people who [say] 'don't have the vaccine, it's got 5G in it and the government are going to follow you and nobody should have it' or you've got the people going 'I've had the vaccine, and everyone should have the vaccine and you are stuffing it up for the rest of us" (Participant 26, Male, 30s)

Echo chambers are discrete and are characterised by a lack of communication across them. In our study we found evidence of a lack of communication between individuals with differing views on vaccines, for example between accepters and refusers. For example, one participant, an accepter, described how having conversations with a family member about vaccines was difficult given the latter's opposing views on Covid:

"I'm happy to get the vaccine . . .but one family member isn't keen on the vaccine, because they are just not convinced of the coronavirus in the first place. . .. I've not had a discussion with them about the vaccine because of his views on coronavirus in general" (Participant 6, Male, 20s)

Similarly, another participant, a refuser, described how she was reluctant to discuss the subject of vaccination with others, particularly those who she knew, or thought, may be in have strongly 'pro-vaccination' views:

"I don't really bring it up now in conversation now with anyone . . . I don't want to get into discussions. . . . I do find there is a lot more of two extreme sides (Participant 8, Female, 40s).

## Discussion

In keeping with the 'continuum of vaccine hesitancy' model [1, 2], we found three main groups of participants, based on their decision or intention to receive a COVID-19 vaccine: Those who had accepted, or were planning on accepting, the vaccine; those who had refused, or were planning on refusing, the vaccine; and those who had not yet decided, or were delaying the decision of, whether or not to get the vaccine. In order to explain these different attitudes, we identified three facilitators (Vaccination as a social norm; Vaccination as a necessity; Trust in science) and six barriers (Preference for "natural immunity"; Concerns over possible side effects; Perceived lack of information; Distrust in government; Conspiracy theories; Covid echo chambers) to vaccine uptake. Although data on actual and eventual vaccine coverage changes rapidly [10], this study provides an in-depth account during the initial roll-out of the

vaccines in the UK (during March-April 2021) into people's decision-making process, including the reasons behind the decision to accept or refuse a vaccine and any hesitancy experienced.

The concept of 'vaccine delay' [1, 2] can therefore be a useful concept (as a form of hesitancy, along with refusal) through which to understand why some are reluctant or unsure as to whether or when they will receive a COVID-19 vaccine, since many of those who were unsure, characterised their decision as one that was ongoing. We found two main reasons for the existence of vaccine delay. Firstly, some were delaying due to a perceived need for more information. Secondly, others were delaying until they were "required" to be vaccinated.

Our findings can be understood in relation to broader conceptual models of vaccine hesitancy, including the 'Five C's' model [3] and the WHO SAGE Working Group on Vaccine Hesitancy Determinants Matrix [2], specifically to the case of COVID-19. Our findings suggest that, as per the 'Five C's' model, particularly Confidence, and Complacency were major factors explaining vaccine hesitancy (concerns over convenience were not apparent in our study, perhaps due to the fact that at the time of data collection, the UK was seeing a rapid and lauded vaccine roll-out via its National Health Service (NHS) (the UK had between January and 22nd April 2021, administered approximately 35 million total doses, with approximately 60% of the total population aged 16 and over having received at least one dose [10]). For example, 'confidence' in the efficacy and safety of the vaccines was a major facilitator or barrier depending on a person's perspective, especially in relation to the extent to which they trusted science or government (and the extent to which they saw the latter as influencing the former) [20]). Also, 'complacency' accounted for why some delayers and refusers were reluctant to be vaccinated. This complacency took the form of a perception of low personal risk and a valuing of "natural immunity" and might be better thought of as a form of 'lay knowledge' or 'medical folk wisdom' [21, 22]. As such, rather than constructing it as 'complacency' per se, it might be more useful to understand how perceptions of low personal risk can be offset by constructing vaccination as a 'collective responsibility–one that even those at relatively low personal risk from serious COVID-19 illness should do to protect others.

The Vaccine Hesitancy Determinants Matric holds that contextual, individual and group, and vaccine-specific issues all impact the extent to which people are accepting of or hesitant towards vaccination [1]. For instance, vaccination as a social norm was found to be an important individual and group influence. We found that a major barrier in the context of COVID-19 is the existence of conspiracy theories and Covid echo chambers. Thus, reducing the circulation and belief in conspiracy theories will likely help control the spread of COVID-19 [23], including in this case through potentially increasing vaccine uptake, perhaps particularly amongst BAME communities. Research suggests that people may be drawn to conspiracy theories when they promise to satisfy epistemic (e.g. desire for certainty), existential (e.g. a desire for control) and social (e.g. a desire to 'fit in' within a group) motives [24]. BAME communities may be particularly at risk from a lack of knowledge and safety, because of their historical marginalization in society and because of the fact that morbidity and mortality from COVID-19 has been higher. Research suggests that experiences of ostracism, including due to an individual's race or ethnicity, may lead to greater belief in conspiracy theories, perhaps as a defence mechanism [24, 25]. Further research on the role of 'Covid echo chambers' is needed. Our findings suggest that some people are reluctant to engage with others who hold, or may hold, differing opinions–particularly since COVID-19 policy is such a divisive and emotionally-charged issue. This lack of communication across echo chambers can have an 'opinion reinforcing' effect [25]. In the context of COVID-19 vaccination, strongly 'pro-vaccination' advertising or opinions may even be having a counterproductive effect for some, encouraging people to 'double-down' on their opposition or adding to their hesitancy. As such, working

with individuals and communities, engaging with rather than criticising or dismissing their concerns (both legitimate and illegitimate) via mutually respectful dialogue is essential [26].

Vaccine delay may be usefully understood in relation to broader conceptual models on 'patient delay' [27]. In the context of COVID-19 vaccines, 'appraisal delay' (or decisional delay) [28] can be thought of as the question of 'should I get vaccinated'? In order to reduce decisional delay, it is particularly important for policy and health organisations to address informational barriers, including some people's perceived lack of information, the existence of conspiracy theories, and the existence of 'Covid echo chambers'. For participants in this study, the lack of information was seen to be partly due to the fact that COVID-19 is a novel disease (i.e. scientists don't yet have enough information) and due to the fact that insufficient or unclear information was being communicated to them (by e.g. medical or political agencies or individuals). Also, some felt that the information around vaccines and their efficacy or safety was at times too complex to understand, especially in light of new developments. This is perhaps another example of a phenomenon identified throughout the pandemic, referred to as 'alert fatigue'. This is where frequently changing information (e.g. policies, guidelines, advisories) becomes increasingly difficult to interpret, comprehend and retain for members of the public [29]. Also, utilization delay (as a form of behavioural delay) can be understood as the time between when a person decides they will need medical intervention (i.e. a vaccine) and the deciding act on that decision (i.e. to get vaccinated). Delayers in this sense had already decided they were going to "need" to be vaccinated. Utilization delay entails the individual asking themselves, is the medical care (i.e. vaccination) worth the costs? In this case, many delayers had decided that the benefits of vaccination–as a 'passport' to international travel—were worth the perceived costs (e.g. the perceived infringement of their right to refuse a vaccine or concerns over potential future side effects). The decision to get vaccinated was perceived as not an entirely voluntary one, but one into which they were being 'nudged' or even indirectly forced—via the assumption that vaccine passports would be required in the future, especially for international travel.

## Limitations

There are a number of limitations to note. Firstly, as with all qualitative studies, the generalizability of the findings is limited. As such, the study cannot be used to make generalisations about how prevalent vaccine acceptance or hesitancy is. Additionally, there was a particularly small number of those in our sample who had refused or were planning to refuse a vaccine and so further research specifically on a larger group of those who have refused the vaccine is needed.

Another limitation of the study is that although attempts were made to recruit and include as diverse a sample as possible, there is a relative underrepresentation of older adults (aged 50 + in the sample). However, for the purposes of the research question around vaccine uptake having a younger sample may be of benefit. Further strengths and limitations of the overall methodology and recruitment in the wider study are discussed in prior publications [10, 11]. A further limitation is that the COVID-19 pandemic has (and at time of writing continues to) evolve rapidly, and so subsequent developments (e.g. new variants), may have affected peoples around vaccine uptake. Further research will explore evolving perceptions and any subsequent decisions. However, a particular strength of this study is its ability to provide in-depth and nuanced context as to the reasons behind vaccine acceptance, delay or refusal in the context of COVID-19 among residents in the UK.

## Conclusion

This study has provided an in-depth examination of the reasons behind participants' decisions around getting their initial COVID-19 vaccines. In order to address vaccine uptake and

vaccine hesitancy, it is useful to understand the reasons behind people's decisions to accept or refuse an offer of a vaccine. This qualitative study has suggested that, in the UK, three facilitators—Vaccination as a social norm; Vaccination as a necessity; Trust in science—and six barriers—Preference for "natural immunity"; Concerns over possible side effects; Distrust in government; Perceived lack of information; Conspiracy theories; Covid echo chambers–can help understand *why* people decide to get a COVID-19 vaccine or not. Future qualitative research looking at different countries is necessary to explore similarities and differences with this study's findings, given the highly contextual nature of vaccine confidence and vaccine hesitancy.

The finding that convenience was not reported as an issue related to vaccine hesitancy in this study is also important to note. While there appears to have been good access to the vaccines among the general population during the study period, it is important for the key messaging to the community to be well thought out and in line with public belief systems, concerns, and potential misinformation. Findings from this study can help direct such messaging for clinicians/medical providers, community leaders, and public health practitioners. The COVID-19 pandemic, and its policy response, has, and will continue, to rapidly evolve. Further research, particularly longitudinal and comparative research, is needed to explore the evolution of attitudes to vaccines as the pandemic continues.

## Author Contributions

**Conceptualization:** Simon N. Williams, Christopher J. Armitage, Kimberly Dienes, John Drury, Tova Tampe.

**Data curation:** Simon N. Williams, Kimberly Dienes.

**Formal analysis:** Simon N. Williams, Kimberly Dienes.

**Funding acquisition:** Simon N. Williams, Christopher J. Armitage, Kimberly Dienes.

**Investigation:** Simon N. Williams, Christopher J. Armitage, Kimberly Dienes.

**Methodology:** Simon N. Williams, Christopher J. Armitage, Kimberly Dienes.

**Project administration:** Simon N. Williams, Kimberly Dienes.

**Resources:** Simon N. Williams.

**Supervision:** John Drury.

**Writing – original draft:** Simon N. Williams.

**Writing – review & editing:** Simon N. Williams, Christopher J. Armitage, Kimberly Dienes, John Drury, Tova Tampe.

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
