## [Decision Letter · Decision Letter 0]

6 Jun 2022

PONE-D-22-08555Public decisions about COVID-19 vaccines: A UK-based qualitative studyPLOS ONE

Dear Dr. Williams,

Thank you for submitting your manuscript to PLOS ONE. After careful consideration, we feel that it has merit but does not fully meet PLOS ONE’s publication criteria as it currently stands. Therefore, we invite you to submit a revised version of the manuscript that addresses the points raised during the review process.

The manuscript in its current form requires major revisions for it to be potentially reconsidered for any further review and without guarantee of acceptance. In addition to the comments provided by the two reviewers, I have provided additional comments that the authors must address. I strongly urge the authors to first discuss among themselves and decide if they believe they can overcome the major concerns that the reviewers and I have outlined. If the authors feel that they can address all these comments adequately, then I’d be happy to reconsider a resubmission.

Big picture: The study aims to address an important public health topic, but its methods have major flaws as currently described. It is unclear if the authors are not accurately describing what they did or if what they did was flawed. For instance, the authors’ operationalization of the WHO SAGE conceptualization of vaccine hesitancy is inaccurate. The authors operationalized vaccine hesitancy as a “delay in vaccination” whereas WHO conceptualizes it as the “refusal or delay” vaccinate despite availability of vaccines. There are drawbacks and criticisms to the WHO SAGE conceptualizations and there are other frameworks/models that have emerged that the authors could have used to inform their work (e.g. 5C Model; BeSD Framework). This may be that the authors are not steeped into the vaccination confidence literature/field, which may explain some of the conceptual / theoretical issues.Methods section – data collection: How exactly was the snowball sampling conducted? It is insufficient to simply mention snowball sampling was done without specifying the details, especially given the online-based application of the sampling.Methods section – analysis: Insufficient to say that the two analysts met and discussed. More important to say what came about from their discussions. For instance, if they had different interpretations, how did they resolve those? What as the role of the other research team members?Methods section – analysis: How was the initial coding done? Deductively or inductively? Specify more details and be clear about how exactly the deductive and inductive coding of the transcript were done.Methods section – analysis: The authors state that “After agreeing on a provisional coding framework, remaining transcripts were coded (with additional themes being added as they emerged).” I do not understand why “themes” would be added at the manuscript coding phase. Are you referring to additional inductive codes that are identified in the coding process? Clarify and explain more clearly.Methods section – analysis: The process that was used to go from codes to themes was quite unclear and without any audit trail to follow. This is a major weakness that must be addressed.Methods section – data collection: It is good practice under qualitative data reporting guidelines (e.g., refer to COREQ) to ensure that the authors state in the manuscript which of the authors had a direct role in the analysis by including their initials in specific areas of the analysis section. Right now, the authors have included vague descriptions that are anonymized.Results - decisions about vaccination: The first paragraph in this section should've been in the analysis section and not in the methods. The sentence doesn’t reflect any findings; instead, it elaborates on the analytical procedures used to categorize the ‘three types’ of vaccination decisions. The finding would be what came out of the analytical structuring of the meaning units on the underlying decisions to vaccinate, delay, or refuse.Results (general): It would be important to briefly describe each of the 6 groups based on their vaccination status to better understand the composition of the groups. Also, in focus group analysis, the analytical unit is the group, not individuals per se; though individuals make up the groups. The ways the results have been quantified based on individual responses (akin to a survey) is inappropriate.Results – vaccine acceptance: Did you put participants into a focus group based on their vaccination status / vaccination intention ahead of time? If so, this should have been stated in the methods. Or was it simply coincidental that the one group of participants had all been vaccinated or planned to vaccinate?General feedback: The writing / grammar in many parts of this manuscript need heavy editing to improve clarity of meaning. Also, it would be helpful to add page numbers and line numbers for ease of providing feedback.

We look forward to receiving your revised manuscript.

Kind regards,

Mohamed F. Jalloh, PhD, MPH

Academic Editor

PLOS ONE

Journal Requirements:

“SW, KD, CJA have been funded by the Manchester University Centre for Health Psychology for this work.

SW received a Greatest Need Fund from Swansea University to csupport this research.“

“CJA is supported by NIHR Manchester Biomedical Research Centre and NIHR Greater Manchester Patient Safety Translational Research Centre.”

Reviewers' comments:

Reviewer's Responses to Questions

**Comments to the Author**

1. Is the manuscript technically sound, and do the data support the conclusions?

Reviewer #1: Partly

Reviewer #2: Yes

2. Has the statistical analysis been performed appropriately and rigorously? 

Reviewer #1: N/A

Reviewer #2: N/A

3. Have the authors made all data underlying the findings in their manuscript fully available?

Reviewer #1: Yes

Reviewer #2: No

4. Is the manuscript presented in an intelligible fashion and written in standard English?

Reviewer #1: No

Reviewer #2: Yes

5. Review Comments to the Author

Reviewer #1: Thank you for sharing this very interesting manuscript that captures a unique timepoint related to COVID-19: vaccine attitudes during early roll-out.

Please find key comments below:

Writing style: At times, the wording is repetitive or not needed (ex. “in the present paper”, “XX”), use of language that infers statistical analyses (respondents ‘more likely to report…” vs. ‘respondents reported’…”), repetitive language in intro, results, and discussion; use of some of the same quotes more than once. Often the preceding or succeeding sentences to a quote were a reiteration of the quote in the authors’ voice rather than writing that weaved together the findings into a narrative. Suggest working with an experienced scientific writer/editor to improve the writing quality.

Introduction: A key piece of information that frames the context of this time was in the discussion section, “at the time of data collection, the UK was seeing a rapid and lauded vaccine roll-out… and 22nd April 2021, administered approximately 35 million total doses, with approximately 60% of the total population aged 16 and over having received at least one dose [7])”, please move the introduction.

Methods: Include interrater reliability of coders. Include as a limitation the small sample size for those that refused vaccination (2 out of the 29 respondents). The discussion guide did not include questions that would help inform policy or programming to help shift attitudes, norms, or uptake.

Results and Discussion: Due to the limited number of respondents (29 divided into three categories [accepters, delayers, refusers]), the depth of findings were limited, and, at times, the same quote was used to make different points. As there was a small sub-sample of ‘refusers’ there was the particularly limited breadth and depth of information gathered to inform vaccine health promotions and interventions to increase uptake/move refusers to delayers to accepters.

The discussion section should be strengthened with the inclusion of scientific literature of behavioral strategies/interventions, health promotion and communication efforts, and policies from previous vaccine-related or health behavior change public health initiatives. Discussion sections are to draw upon the findings of the study (vs. repeating the findings) to propose the next steps (interventions, communication, and policy) informed/guided by the literature of similar situations.

Reviewer #2: In general, the study presented in this manuscript is technically sound and offers important insights. I would suggest the following edits/revisions to further improve the manuscript.

Methods

*It would be helpful to clarify if the focus groups were recorded and transcribed.

*I suggest adding a description of how the participants were segmented into the three "types" discussed in first few pages of the results. It's mentioned that the continuum of vaccine hesitancy was used for the analysis of vaccine uptake barriers and facilitators, but not entirely clear how that led to classification of each participant.

Results

*It's somewhat misleading to quantify the three segments e.g., vaccine acceptance (n=15 (52%). Since this is a qualitative study, the size of each segment is not as relevant and some readers may incorrectly associate larger size with higher degree of validity, etc. I suggest removing these numbers.

*Discussion around the Black and Asian Minority Ethnic (BAME) participants in the "conspiracy theories and misinformation" section appears to be important, but feels a bit incongruous because no context is given in the introduction. I suggest adding some literature on vaccine equity or racial/ethnic disparities and introduce the concept earlier if it comes up in results.

*In general, the results section feels a little too long. Wherever possible, I suggest condensing some of the sub-sections, perhaps by paraphrasing participant quotes.

Discussion/Conclusion

*Some of the theoretical frameworks referenced are somewhat outdated e.g., 3 Cs. I suggest also looking at are more recent frameworks like the BeSD model (https://www.who.int/publications/i/item/WHO-2019-nCoV-vaccination-demand-planning-2021.1#new_tab)

*I would suggest contextualizing "vaccine hesitancy" within the broader discussions of social and behavioral barriers to vaccine uptake. Hesitancy refers to specific individual-level perceptions and attitudes (which I understand is the focus of this paper), but addressing hesitancy alone would not be sufficient in many settings because of other practical or environmental factors. It's mentioned that issues related to convenience did not come up in the focus groups, but it would still be helpful to discuss in more depth why that was the case and if there are other studies that suggest otherwise.

*I would consider rephrasing the following sentence "As such, those working in public health in the UK and

comparable countries (global inequalities with vaccine access notwithstanding) might benefit

from incorporating the three facilitators.". Since this is a qualitative study and also because vaccine confidence/demand/hesitancy is usually highly contextual, I am unsure if such a general claim can be made.

6. PLOS authors have the option to publish the peer review history of their article (what does this mean?). If published, this will include your full peer review and any attached files.

Reviewer #1: No

Reviewer #2: **Yes: **Atsuyoshi Ishizumi

---

## [Author Response · Author response to Decision Letter 0]

12 Sep 2022

Dear Editorial team and reviewers, 

we are extremely grateful for your review - we think this has strengthened the manuscript significantly. Please see below our point-by-point major revisions (also in the response letter in red italics) Thanks and kind regards

Simon Williams (on behalf of the authors)

1. Big picture: The study aims to address an important public health topic, but its methods have major flaws as currently described. It is unclear if the authors are not accurately describing what they did or if what they did was flawed. For instance, the authors’ operationalization of the WHO SAGE conceptualization of vaccine hesitancy is inaccurate. The authors operationalized vaccine hesitancy as a “delay in vaccination” whereas WHO conceptualizes it as the “refusal or delay” vaccinate despite availability of vaccines. There are drawbacks and criticisms to the WHO SAGE conceptualizations and there are other frameworks/models that have emerged that the authors could have used to inform their work (e.g. 5C Model; BeSD Framework). This may be that the authors are not steeped into the vaccination confidence literature/field, which may explain some of the conceptual / theoretical issues.

Many thanks to the reviewers and editorial team for these comments. We strongly believe that the methodology itself was not flawed, but agree we did not fully and in sufficient detail, with sufficient accuracy, describe the methods in our first submission. We believe that the very helpful comments have been addressed and we have now added more details and clarifications to the below points to elucidate the methods employed.

We have emphasised the initial definition of vaccine hesitancy as delay or refusal: “Vaccine hesitancy can be defined as “the delay in acceptance or refusal of vaccination despite availability of vaccination services”.” (p,3). Also we have clarified in the discussion that vaccine delay is a particular form of hesitancy: “The concept of ‘vaccine delay’ [1] [2] can therefore be a useful concept (as a form of hesitancy, along with refusal) through which to understand why some are reluctant or unsure as to whether or when they will receive a COVID-19 vaccine, since many of those who were unsure, characterised their decision as one that was ongoing.” 

We thank the reviewers for suggesting the inclusion of the more recent frameworks – we have incorporated discussion of the 5Cs and BEsD frameworks 

2. Methods section – data collection: How exactly was the snowball sampling conducted? It is insufficient to simply mention snowball sampling was done without specifying the details, especially given the online-based application of the sampling.

We have added additional detail about the sampling procedure: “Recruitment for the study took place primarily via non-probability, opportunity sampling. Recruitment included using social media advertising (Facebook ads and via posting ads on Twitter), other online advertising (e.g. online ‘free-ads’ such as Gumtree), as well as snowball sampling (e.g. asking participants who had taken part in a focus group to distribute the study ad to others they felt might be interested in participating). “

3. Methods section – analysis: Insufficient to say that the two analysts met and discussed. More important to say what came about from their discussions. For instance, if they had different interpretations, how did they resolve those? What as the role of the other research team members?

Much more detail has been added here., eg: “Two authors (SW and KD) primarily analysed the data, in consultation with the other three authors (CA, JD, TT). The first stage of FA is data familiarisation. SW and KD independents read three focus group transcripts initially, and independently assigned preliminary codes to specific lines or chunks of texts. The two authors met regularly (after each transcript) to identify commonalities and differences in their coding and discussed frequent or significant codes which were beginning to form emergent themes – i.e. patterns in the data, codes clustering around a common concept (e.g. ‘preference for natural immunity’). To ensure inter-coder reliability, we followed Miles and Huberman’s rule of thumb – ensuring that coding agreement was above 80% [15] [16]. Coding differences were resolved through ‘negotiated agreement’ [17].”

4. Methods section – analysis: How was the initial coding done? Deductively or inductively? Specify more details and be clear about how exactly the deductive and inductive coding of the transcript were done.

We have added more detail about the use of Framework Analysis – including how we combined elements of deductive and inductive coding using this flexible approach.

5. Methods section – analysis: The authors state that “After agreeing on a provisional coding framework, remaining transcripts were coded (with additional themes being added as they emerged).” I do not understand why “themes” would be added at the manuscript coding phase. Are you referring to additional inductive codes that are identified in the coding process? Clarify and explain more clearly.

We have edited and clarified and added detail throughout the analysis section to explain the analysis process in more depth – e.g. “The two authors met regularly (after each transcript) to identify commonalities and differences in their coding and discussed frequent or significant codes which were beginning to form emergent themes – i.e. patterns in the data, codes clustering around a common concept (e.g. ‘preference for natural immunity’).” We have been careful to edit so as not to conflate the terms codes with themes.

6. Methods section – analysis: The process that was used to go from codes to themes was quite unclear and without any audit trail to follow. This is a major weakness that must be addressed.

We have added much more detail here. (e.g.” to identify commonalities and differences in their coding and discussed frequent or significant codes which were beginning to form emergent themes – i.e. patterns in the data, codes clustering around a common concept (e.g. ‘preference for natural immunity’)”.

7. Methods section – data collection: It is good practice under qualitative data reporting guidelines (e.g., refer to COREQ) to ensure that the authors state in the manuscript which of the authors had a direct role in the analysis by including their initials in specific areas of the analysis section. Right now, the authors have included vague descriptions that are anonymized.

This detail has been added: e.g. “The other three authors (CA, JD, TT) were not involved primarily in the data analysis but were consulted throughout the FA – and provided input into code definition and theme generation, they also provided comments which were used to resolve any disagreements between SW and KD’s coding.

8. Results - decisions about vaccination: The first paragraph in this section should've been in the analysis section and not in the methods. The sentence doesn’t reflect any findings; instead, it elaborates on the analytical procedures used to categorize the ‘three types’ of vaccination decisions. The finding would be what came out of the analytical structuring of the meaning units on the underlying decisions to vaccinate, delay, or refuse.

We have moved this paragraph to the methods (analysis) section

9. Results (general): It would be important to briefly describe each of the 6 groups based on their vaccination status to better understand the composition of the groups. Also, in focus group analysis, the analytical unit is the group, not individuals per se; though individuals make up the groups. The ways the results have been quantified based on individual responses (akin to a survey) is inappropriate.

We have removed the numbers/frequencies based on this recommendation, to avoid quasi-quantification and to focus on qualitative views and meanings. 

We have also clarified that focus group composition was carried over from previous rounds of the study earlier in the pandemic, and that as a result groups were not formed based on vaccine status:

“Focus groups were not arranged according to any pre-existing views or decisions around vaccinations (i.e. we did not purposefully put those who were not intending on getting vaccinated in the same group for example). This was largely because of the longitudinal nature of the overall study, and our initial decision to try and keep the membership of each focus group the same over time in order to build rapport, familiarity and openness within the groups (we did not collect data on vaccine views during the initial recruitment and group allocation in March 2020). Each group contained a mixture of those who had already received at least one vaccine (n=15) and those who had either already refused a vaccine or who were delaying their decision to get a vaccine (n=14). In March 2021, participants were invited to take part in a rapid round of focus groups on the topic of vaccines. …”

10. Results – vaccine acceptance: Did you put participants into a focus group based on their vaccination status / vaccination intention ahead of time? If so, this should have been stated in the methods. Or was it simply coincidental that the one group of participants had all been vaccinated or planned to vaccinate?

We have added detail to explain how groups were formed at the start of the study and not based on vaccine intention.:

“Each group contained a mixture of those who had already received at least one vaccine (n=15) and those who had either already refused a vaccine or who were delaying their decision to get a vaccine (n=14). Although this was primarily due to the groups being previously formed. One potential limitation of this focus group composition was that those who were refusing or delaying vaccination might have felt less comfortable expressing their opinion (since, generally getting vaccinated was seen by many as a social norm - see below). However, the focus group facilitator sought to ensure that all participants felt comfortable expressing their views, and that dialogue within the group was not hostile and as respectful and open as possible. Also, questions were phrased in non-leading, non-judgemental a way as possible.”

11. General feedback: The writing / grammar in many parts of this manuscript need heavy editing to improve clarity of meaning. Also, it would be helpful to add page numbers and line numbers for ease of providing feedback.

We have thoroughly edited, reviewed and proofed the revised manuscript and have focused on improving readability while doing so. Page and line numbers have been added (apologies for not adding first time round).

We look forward to receiving your revised manuscript.

Kind regards,

Mohamed F. Jalloh, PhD, MPH

Academic Editor

PLOS ONE

Journal Requirements:

“SW, KD, CJA have been funded by the Manchester University Centre for Health Psychology for this work.

SW received a Greatest Need Fund from Swansea University to csupport this research.“

We havea dded this: "The funders had no role in study design, data collection and analysis, decision to publish, or preparation of the manuscript."

“CJA is supported by NIHR Manchester Biomedical Research Centre and NIHR Greater Manchester Patient Safety Translational Research Centre.”

We have added the following to the revised cover letter:

This does not alter our adherence to PLOS ONE policies on sharing data and materials.

We have added the following to the revised cover letter:

This does not alter our adherence to PLOS ONE policies on sharing data and materials.

We have included as follows

As per the PLoS One policy on data collected as part of qualitative research, we have made excerpts of the transcripts relevant to the study available upon request.

For any discussions about the data set please contact the relevant members of the research team Dr Simon Williams (s.n.williams@swansea.ac.uk) or Dr Kimberly Dienes k.dienes@swansea.ac.uk.

This is only mentioned in the methods section

Reviewers' comments:

Reviewer's Responses to Questions

Comments to the Author

1. Is the manuscript technically sound, and do the data support the conclusions?

Reviewer #1: Partly

Reviewer #2: Yes

2. Has the statistical analysis been performed appropriately and rigorously?

Reviewer #1: N/A

Reviewer #2: N/A

3. Have the authors made all data underlying the findings in their manuscript fully available?

Reviewer #1: Yes

Reviewer #2: No

4. Is the manuscript presented in an intelligible fashion and written in standard English?

Reviewer #1: No

Reviewer #2: Yes

5. Review Comments to the Author

Reviewer #1: Thank you for sharing this very interesting manuscript that captures a unique timepoint related to COVID-19: vaccine attitudes during early roll-out.

Thank you for your very helpful review

Please find key comments below:

Writing style: At times, the wording is repetitive or not needed (ex. “in the present paper”, “XX”), use of language that infers statistical analyses (respondents ‘more likely to report…” vs. ‘respondents reported’…”), repetitive language in intro, results, and discussion; use of some of the same quotes more than once. Often the preceding or succeeding sentences to a quote were a reiteration of the quote in the authors’ voice rather than writing that weaved together the findings into a narrative. Suggest working with an experienced scientific writer/editor to improve the writing quality.

We have thoroughly edited, reviewed and proofed the revised manuscript and have focused on improving readability while doing so.

Introduction: A key piece of information that frames the context of this time was in the discussion section, “at the time of data collection, the UK was seeing a rapid and lauded vaccine roll-out… and 22nd April 2021, administered approximately 35 million total doses, with approximately 60% of the total population aged 16 and over having received at least one dose [7])”, please move the introduction.

Thanks – we have integrated this into the introduction where we discuss the roll out. (i.e.: “For context, during this period the UK was experiencing a rapid roll out of COVID-19 vaccines (Astra Zeneca and Pfizer-BioNTech) via the National Health Service, with first doses going from approximately one-third to one-fifth of the eligible adult population, and second doses going from approximately 2% to 17% of the eligible adult population, with doses being prioritised amongst older adults and those with certain underlying health conditions (clinically vulnerable and clinically extremely vulnerable adults)”)

Methods: Include interrater reliability of coders. Include as a limitation the small sample size for those that refused vaccination (2 out of the 29 respondents). The discussion guide did not include questions that would help inform policy or programming to help shift attitudes, norms, or uptake.

We did not run formal IRR/ICR statistics, but we have explained in more detail how we coded independently and checked consistency and resolved disagreement, following Miles and Huberman: “Two authors (SW and KD) primarily analysed the data, in consultation with the other three authors (CA, JD, TT). The first stage of FA is data familiarisation. SW and KD independents read three focus group transcripts initially, and independently assigned preliminary codes to specific lines or chunks of texts. The two authors met regularly (after each transcript) to identify commonalities and differences in their coding and discussed frequent or significant codes which were beginning to form emergent themes – i.e. patterns in the data, codes clustering around a common concept (e.g. ‘preference for natural immunity’). To ensure inter-coder reliability, we followed Miles and Huberman’s rule of thumb – ensuring that coding agreement was above 80% [15] [16]. Coding differences were resolved through ‘negotiated agreement’ [17]

We have added additional detail to the limitations section: “Additionally, there was a particularly small number of those in our sample who had refused or were planning to refuse a vaccine and so further research specifically on a larger group of those who have refused the vaccine is needed.”

We have added the fact we did not have questions specifically related to policy or programming on our focus group schedule this to the limitations section. However - we have also added detail that the aim of the exploratory study was to understand attitudes to a novel vaccine (e.g. how/why people were worried over potential side effects or what people felt about vaccine passports) – with a view to subsequent research or policymakers or practitioners translating this research to inform their interventions or communications.

Results and Discussion: Due to the limited number of respondents (29 divided into three categories [accepters, delayers, refusers]), the depth of findings were limited, and, at times, the same quote was used to make different points. As there was a small sub-sample of ‘refusers’ there was the particularly limited breadth and depth of information gathered to inform vaccine health promotions and interventions to increase uptake/move refusers to delayers to accepters.

This has been acknowledged in the limitations section. We have also added a sentence to note that some of the facilitators could be harnessed or emphasised in order to help encourage uptake (e.g. promoting vaccination as a social norm through various online and offline networks and media).

The discussion section should be strengthened with the inclusion of scientific literature of behavioral strategies/interventions, health promotion and communication efforts, and policies from previous vaccine-related or health behavior change public health initiatives. Discussion sections are to draw upon the findings of the study (vs. repeating the findings) to propose the next steps (interventions, communication, and policy) informed/guided by the literature of similar situations.

Reviewer #2: In general, the study presented in this manuscript is technically sound and offers important insights. I would suggest the following edits/revisions to further improve the manuscript.

Thank you for your very helpful review.

Methods

*It would be helpful to clarify if the focus groups were recorded and transcribed.

*I suggest adding a description of how the participants were segmented into the three "types" discussed in first few pages of the results. It's mentioned that the continuum of vaccine hesitancy was used for the analysis of vaccine uptake barriers and facilitators, but not entirely clear how that led to classification of each participant.

We have clarified this with the following sentence added: Focus groups were recoded and transcribed. 

We have added more detail to explain how we coded into categories, e.g.:

“Using the vaccine hesitancy continuum model [1], we analysed data to look for, and explore, three main types of vaccine decisions in the data: those who had accepted, or were planning on accepting, the vaccine; those who had refused, or were planning on refusing, the vaccine; and those who had not yet decided, or were delaying the decision of, whether or not to get the vaccine. As such, each participant was coded into one of these three main categories based on their decision or intention (e.g. statements that indicated they were not sure if they wanted a vaccine were coded into three main categories: accept, delay, refuse. Sub-coding sought to bring out the nuances within these simplified intention/decision types (e.g. “accept but unsure”).” 

Results

*It's somewhat misleading to quantify the three segments e.g., vaccine acceptance (n=15 (52%). Since this is a qualitative study, the size of each segment is not as relevant and some readers may incorrectly associate larger size with higher degree of validity, etc. I suggest removing these numbers.

We have removed these numbers and percentages instead focusing on qualitative themes.

*Discussion around the Black and Asian Minority Ethnic (BAME) participants in the "conspiracy theories and misinformation" section appears to be important, but feels a bit incongruous because no context is given in the introduction. I suggest adding some literature on vaccine equity or racial/ethnic disparities and introduce the concept earlier if it comes up in results.

We have included a sentence on BAME inequalities in vaccine hesitancy the introduction to introduce this :

“Evidence also suggests that there a number of inequalities in vaccine uptake, including in the UK, by ethnicity (where for example some BlackandAsian Minority Ethnic groups have significantly lower vaccine uptake and significantly higher vaccine hesitancy”

We have also elaborated on the discussion in the discussion section: 

“Thus, reducing the circulation and belief in conspiracy theories will likely help control the spread of COVID-19, [23] including in this case through potentially increasing vaccine uptake, perhaps particularly amongst BAME communities. Research suggests that people may be drawn to conspiracy theories when they promise to satisfy epistemic (e.g. desire for certainty), existential (e.g. a desire for control) and social (e.g. a desire to ‘fit in’ within a group) motives. [24] BAME communities may be particularly at risk from a lack of knowledge and safety, because of their historical marginalization in society and because of the fact that morbidity and mortality from COVID-19 has been higher. Research suggests that experiences of ostracism, including due to an individual’s race or ethnicity, may lead to greater belief in conspiracy theories, perhaps as a defence mechanism”

*In general, the results section feels a little too long. Wherever possible, I suggest condensing some of the sub-sections, perhaps by paraphrasing participant quotes.

We have edited this section and tried to reduce quote length as much as possible without losing important meaning/content.

Discussion/Conclusion

*Some of the theoretical frameworks referenced are somewhat outdated e.g., 3 Cs. I suggest also looking at are more recent frameworks like the BeSD model (https://www.who.int/publications/i/item/WHO-2019-nCoV-vaccination-demand-planning-2021.1#new_tab)

We have incorporated the 5C’s and BeSD model into the paper. Thanks 

*I would suggest contextualizing "vaccine hesitancy" within the broader discussions of social and behavioral barriers to vaccine uptake. Hesitancy refers to specific individual-level perceptions and attitudes (which I understand is the focus of this paper), but addressing hesitancy alone would not be sufficient in many settings because of other practical or environmental factors. It's mentioned that issues related to convenience did not come up in the focus groups, but it would still be helpful to discuss in more depth why that was the case and if there are other studies that suggest otherwise.

We have added more detail to try and acknowledge that these individual decisions and perceptions take place within, and ae shaped by social-cultural-political factors, whilst as you say emphasising the purpose of the research was to focus on individual attitudes/decisions within it.

*I would consider rephrasing the following sentence "As such, those working in public health in the UK and

comparable countries (global inequalities with vaccine access notwithstanding) might benefit

from incorporating the three facilitators.". Since this is a qualitative study and also because vaccine confidence/demand/hesitancy is usually highly contextual, I am unsure if such a general claim can be made.

We agree – and have removed this as a conclusion.

---

## [Decision Letter · Decision Letter 1]

26 Oct 2022

Public decisions about COVID-19 vaccines: A UK-based qualitative study

PONE-D-22-08555R1

Dear Dr. Williams,

We’re pleased to inform you that your manuscript has been judged scientifically suitable for publication and will be formally accepted for publication once it meets all outstanding technical requirements.

Kind regards,

Mohamed F. Jalloh, PhD, MPH

Academic Editor

PLOS ONE

Additional Editor Comments (optional):

Reviewers' comments:

Reviewer's Responses to Questions

**Comments to the Author**

1. If the authors have adequately addressed your comments raised in a previous round of review and you feel that this manuscript is now acceptable for publication, you may indicate that here to bypass the “Comments to the Author” section, enter your conflict of interest statement in the “Confidential to Editor” section, and submit your "Accept" recommendation.

Reviewer #2: All comments have been addressed

2. Is the manuscript technically sound, and do the data support the conclusions?

Reviewer #2: Yes

3. Has the statistical analysis been performed appropriately and rigorously? 

Reviewer #2: N/A

4. Have the authors made all data underlying the findings in their manuscript fully available?

Reviewer #2: Yes

5. Is the manuscript presented in an intelligible fashion and written in standard English?

Reviewer #2: Yes

6. Review Comments to the Author

Reviewer #2: (No Response)

7. PLOS authors have the option to publish the peer review history of their article (what does this mean?). If published, this will include your full peer review and any attached files.

Reviewer #2: **Yes: **Atsuyoshi Ishizumi

---

## [Editor Report · Acceptance letter]

14 Nov 2022

PONE-D-22-08555R1 

Public decisions about COVID-19 vaccines: A UK-based qualitative study 

Dear Dr. Williams:

I'm pleased to inform you that your manuscript has been deemed suitable for publication in PLOS ONE. Congratulations! Your manuscript is now with our production department. 

Kind regards, 

on behalf of

Dr. Mohamed F. Jalloh 

Academic Editor

PLOS ONE